# Facial Expressions Recognition for Human–Robot Interaction Using Deep Convolutional Neural Networks with Rectified Adam Optimizer

**DOI:** 10.3390/s20082393

**Published:** 2020-04-23

**Authors:** Daniel Octavian Melinte, Luige Vladareanu

**Affiliations:** Department of Robotics and Mechatronics, Romanian Academy Institute of Solid Mechanics, 010141 Bucharest, Romania; octavian.melinte@imsar.ro

**Keywords:** computer vision, deep learning, convolutional neural networks, advanced intelligent control, facial emotion recognition, face recognition, NAO robot

## Abstract

The interaction between humans and an NAO robot using deep convolutional neural networks (CNN) is presented in this paper based on an innovative end-to-end pipeline method that applies two optimized CNNs, one for face recognition (FR) and another one for the facial expression recognition (FER) in order to obtain real-time inference speed for the entire process. Two different models for FR are considered, one known to be very accurate, but has low inference speed (faster region-based convolutional neural network), and one that is not as accurate but has high inference speed (single shot detector convolutional neural network). For emotion recognition transfer learning and fine-tuning of three CNN models (VGG, Inception V3 and ResNet) has been used. The overall results show that single shot detector convolutional neural network (SSD CNN) and faster region-based convolutional neural network (Faster R-CNN) models for face detection share almost the same accuracy: 97.8% for Faster R-CNN on PASCAL visual object classes (PASCAL VOCs) evaluation metrics and 97.42% for SSD Inception. In terms of FER, ResNet obtained the highest training accuracy (90.14%), while the visual geometry group (VGG) network had 87% accuracy and Inception V3 reached 81%. The results show improvements over 10% when using two serialized CNN, instead of using only the FER CNN, while the recent optimization model, called rectified adaptive moment optimization (RAdam), lead to a better generalization and accuracy improvement of 3%-4% on each emotion recognition CNN.

## 1. Introduction

Humans use their facial expressions to show their emotional states. In order to achieve an accurate communication between humans and robots, the robot needs to understand the facial expression of the person who it is interacting with. The aim of the paper is to develop an end-to-end pipeline for the interaction between a human and NAO robot using computer vision based on deep convolutional neural networks. The paper focuses on enhancing the performance of different types of convolutional neural networks (CNN), in terms of accuracy, generalization and inference speed, using several optimization methods (including the state-of-the-art rectified Adam), FER2013 database augmentation with images from other databases, asynchronous threading at inference time using the Neural Compute Stick 2 preprocessor, in order to develop a straightforward pipeline for emotion recognition on robot applications, mainly NAO robot. Thereby, we are using the pipeline to first localize and align one face from the input image using a CNN face detector for facial recognition as a preprocessing tool for emotion recognition. Then, the proposal from the face recognition (FR) CNN is used as input for the second CNN, which is responsible for the actual facial emotion recognition (FER CNN). The performance of this serialized-CNN model, hereafter referred to as NAO-SCNN, depends on many factors such as the number of images in the training dataset, data augmentation, the CNN architecture, loss function, hyperparameters adjusting, transfer learning, fine-tuning, evaluation metrics, etc., which leads to a complex set of actions in order to develop an accurate and real-time pipeline. The interaction between human and robot can be regarded as a closed loop interaction: the images captured by the robot are first preprocessed by the FR CNN module. The output of this module is a set of aligned face proposals, which are transferred to FER CNN. This second module is responsible for emotion recognition. There are seven emotions that are considered: happiness, surprise, neutral, sadness, fear, angriness and disgust. Based on the emotion detected from the images/videos the robot will adjust its behavior according to a set of instructions carried out using medical feedback from psychologists, physicians, educators and therapists in order to reduce frustration and anxiety through communication with a humanoid robot. In case there is no change in the user emotion, the robot will perform another set of actions Since, at this time, the research is focused on the artificial intelligence solution for emotion recognition the set of actions for the robot interaction is basic (the robot displays the user emotion using video, audio and posture feedback) and the conclusions will regard this matter. Based on the results of this study future social studies regarding the behavior of the persons the robot is interacting with will be developed. The expression recognition has been carried out on NAO robot using one Viola–Jones based CNN trained on AffectNet in order to detect facial expressions in children using the NAO robot with a test accuracy of 44.88% [1]. Additionally, a dynamic Bayesian mixture model classifier has been used for FER human interaction with NAO achieving an overall accuracy of 85% on the Karolinska Directed Emotional Faces (KDEF) dataset [2].

In terms of facial emotion recognition, the recent studies focused on developing and achieving neural models with high accuracy using deep learning. For static images different approaches have been developed and involved: pretraining and fine-tuning, adding auxiliary blocks and layers, multitask networks, cascaded networks and generative adversarial networks. In [3] a multitasking network has been developed in order to predict the interpersonal relation between individuals. It aims for high level interpersonal relation traits, such as friendliness, warmth and dominance for faces that coexist in an image. For dynamic image sequences there are other techniques: frame aggregation, expression intensity network and deep spatiotemporal FER network. Zhao et al. proposed a novel peak-piloted deep network that uses a sample with peak expression (easy sample) to supervise the intermediate feature responses for a sample of non-peak expression (hard sample) of the same type and from the same subject. The expression evolving process from non-peak expression to peak expression can thus be implicitly embedded in the network to achieve the invariance to expression intensities [4]. Fine-tuning works for pretrained networks on datasets that include face or human images but the face information remains dominant. The inference will focus on detecting faces rather than detecting expressions. H. Ding et al. proposed a new FER model that deals with this drawback. The method implies two learning stages. In the first training stage the emotion net convolutional layers are unfrozen, while the face net layers are frozen and provide supervision for emotion net. The fully connected layers are trained in the second stage with expression information [5]. In [6] a two stage fine-tuning is presented. A CNN pretrained on an ImageNet database is fine-tuned on datasets relevant to facial expressions (FER2013), followed by a fine-tuning using EmotiW. To avoid variations introduced by personal attributes a novel identity-aware convolutional neural network (IACNN) is proposed in [7]. An expression-sensitive contrastive loss was also developed in order to measure the expression similarity. There are studies that research other aspects of emotion recognition, such as annotation errors and bias, which are inevitable among different facial expression datasets due to the subjectivity of annotating facial expressions. An inconsistent pseudo annotations to latent truth (IPA2LT) framework has been developed by Jiabei Zeng et al. to train a FER model from multiple inconsistently labeled datasets and large scale unlabeled data [8]. A novel transformations of image intensities to 3D spaces was designed to simplify the problem domain by removing image illumination variations [9]. In [10] a novel deep CNN for automatically recognizing facial expressions is presented with state-of-the art results, while in [11] a new feature loss is developed in order to distinguish between similar features.

There are different types of databases for FER such as Japanese Female Facial Expressions (JAFFE) [12], Extended Cohn–Kanade (CK+) [13], FER 2013 [14], AffectNet [15], MMI (MMI, 2017) [16,17], AFEW [18] and Karolinska Directed Emotional Faces (KDEF) [19].

The CK+ is the most used dataset for emotion detection. The performance of CNN models trained on this dataset are greater than 95% due to the fact that the images are captured in a controlled environment (lab) and the emotions are overacted. The best performance was reached by Yang et al. with a performance of 97.3% [20]. The dataset contains 593 video frames from 123 subjects. The frames vary from 10 to 60 and display a translation from neutral to one of the six desired emotion: anger, contempt, disgust, fear, happiness, sadness and surprise. Not all the subjects provide a video sequence for each emotion. The images in the dataset are not divided into training, validation and testing so it is not possible to perform a standardized evaluation.

The Jaffe is a laboratory-controlled dataset with fewer video frames than CK+. The CNN models tested using this database have performances over 90%, with Hamester et al. reaching 95.8% [21]. The KDEF is also a laboratory-controlled database that focuses on the emotion recognition from five different camera angles. The FER2013 is the second widely used dataset after CK+. The difference is that FER 2013 images are taken from the web and are not laboratory-controlled, the expressions are not exaggerated, thus harder to recognize. There are 28,079 training images, 3589 validation images and 3589 test images that belong to seven classes. The performances using this dataset do not exceed 75.2%. This accuracy has been achieved by Pramerdorfer and Kampel [22], while other researches reached 71.2% using linear support vector machines [23], 73.73% by fusing aligned and non-aligned face information for automatic affect recognition in the wild [24], 70.66% using an end-to-end deep learning framework, based on the attentional convolutional network [25], 71.91% using three subnetworks with different depths [26] and 73.4 using a hybrid CNN–scale invariant feature transform aggregator [27].

The research carried out in this paper aims to improve the accuracy of FER 2013 based CNN models by adding laboratory controlled images from CK+, JaFFE and KDEF. Hopefully there is an emotion compilation that satisfies our requirements and is available on Kaggle [28]. This dataset is divided in 24,336 training, 6957 validation and 3479 test sets. The FER2013 images make up the biggest part of the dataset, around 90% for each emotion.

When it comes to face detection the backbone networks such as VGGNet [29], Inception by Google [30] or ResNet [31] play an important role. In [32] the DeepFace architecture, a nine-layer CNN with several locally connected layers, is proposed for face detection. The authors reported an accuracy of 97.35% on the labeled faces in the Wild dataset. The FaceNet model uses an Inception CNN as the backbone network in order to obtain an accuracy of 99.63% on the widely used labeled faces in the Wild dataset [33]. Another widely used FR network is the VGGFace, which uses a VGGNet base model trained on data collected from the Internet. The network is then fine-tuned with a triplet loss function similar to FaceNet and obtains an accuracy of 98.95% [34]. In 2017, SphereFace [35] used a 64-layer ResNet architecture and proposed an angular softmax loss that enables CNNs to learn angularly discriminative features with an accuracy of 99.42%.

In terms of the human–robot interaction, the research is concentrated on different levels and implies, among others, the grasping configurations of robot dexterous hands using Dezert–Smarandache theory (DSmT) decision-making algorithms [36], developing advanced intelligent control systems for the upper limb [37,38,39,40,41] or other artificial intelligence techniques such as neutrosophic logic [36,41,42], extenics control [43,44] and fuzzy dynamic modeling [45,46,47], applicable for the human–robot interaction with feedback through facial expressions recognition. The research in this paper is focused on developing an AI based computer vision system to achieve the communication human–NAO robot aiming the future researches to develop an end-to-end pipeline system with advanced intelligent control using feedback from the interaction between a human and NAO robot.

## 2. Methods

Direct training of deep networks on relatively small facial datasets is prone to over fitting. Public databases are no larger than 100k images. To compensate this, many studies used data preprocessing or augmentation and additional task-oriented data to pretrain their self-built networks from scratch or fine-tuned on well-known pretrained models. 

In order to choose the appropriate solution for the human–NAO robot face detection two different types of object detectors have been taken into account: single shot detector (SSD) and regional proposal network (RPN). These two architectures have different outputs in practice. The SSD architecture is a small CNN network (around 3 million parameters) with good accuracy for devices with limited computation power. The inference time is very small compared to other large object detectors. The faster region-based convolutional neural network (Faster R-CNN), instead, has high accuracy, better results on small objects, but the inference time takes longer than SSD models and is not suitable for real time applications with low computational power.

The FR architecture consists of a base model (Inception for SSD architecture and ResNet + Inception for Faster R-CNN) that is extended to generate multiscale or regional proposals feature maps. These models were pretrained on different databases: the SSD+InceptionV2 was trained on the common objects in context (COCO) dataset, while the Faster R-CNN was pretrained on a much larger database, the Google OpenImage database. 

### 2.1. Face Detection CNN 

The human faces share a similar shape and texture, the representation learned from a representative proportion of faces can generalize well to detect the others, which are not used in the network training process. The performance of the trained model depends on many factors such as number of images in the training dataset, data augmentation, the CNN architecture, loss function, hyper-parameters adjusting, transfer learning, fine-tuning, evaluation metrics, etc., which leads to a complex set of actions in order to develop the entire pipeline.

The face detector was used to localize faces in images and to align them to normalized coordinates afterwards. The CNN architecture is made up of a backbone network, a localization CNN and the fully connected layers for classification. The backbone networks used for facial recognition were the Inception V2 for SSD architecture and ResNet-InceptionV2 for Faster R-CNN, respectively. In terms of the localization network there were two types of architecture: SSD and Faster R-CNN. The SSD architecture adds auxiliary CNNs after the backbone network, while the Faster-RCNN uses a regional proposal network (RPN) for proposing regions in the images, which were further sent to the CNN, which was also used as a backbone. The classification layers of both face detectors were at the top of the architecture and used the softmax loss function together with L2 normalization in order to adjust the localization of the face region and to classify the image.

#### 2.1.1. Inception Based SSD Model

The SSD architecture uses an Inception V2 pretrained model as a backbone. The Inception model was designed for solving high variation in the location of information, thus is useful for localization and object detection when is added as base network for SSD and RPN architectures. Different types of kernels with multiple sizes (7 × 7, 5 × 5 and 3 × 3) were used. Larger kernels can look for the information that is distributed more globally, as the smaller one search in the information that is not as sparse. There is one important filter that is also used for this type of CNN, which is the 1 × 1 convolution for reducing or increasing the number of feature maps. The network is 22 layers deep when counting only layers with parameters (or 27 layers with pooling). The overall number of layers (independent building blocks) used for the construction of the network is about 100. The SSD architecture adds auxiliary structure to the network to produce multiscale feature maps and convolutional predictors for detection. At prediction time, the network generates scores for the presence of each object category in each default box and produces adjustments to the box to better match the object shape. Additionally, the network combines predictions from multiple feature maps with different resolutions to naturally handle objects of various sizes [48].

#### 2.1.2. Inception-ResNetV2 Based Faster R-CNN Model

The face detection and alignment is tested with a second object detector, the Faster R-CNN with Inception and ResNet as backbone CNN. Faster R-CNN convolutional networks use regional proposal method to detect the class and localization of the objects. The architecture is made up of two modules: the regional proposal network (RPN) and the detection network. The RPN has CNN architecture with anchors and multiple region proposals at each anchor location to output a set of bounding boxes proposals with a detection score. The detection network is a Fast R-CNN network that uses the region proposals from RPN to search for objects in those regions of interest (ROI). ROI pooling was performed and then resulted feature maps pass through CNN and two fully connected layers for classification (Softmax) and bounding box regression. The RPN and the detection network share a common set of convolutional layers in order to reduce computation [49].

As comparison, it is a common fact that the Faster R-CNN performs much better when it comes to detecting small object, but shares the same accuracy with the SSD network when detecting large objects. Due to the complex architecture of Faster R-CNN, the inference time is three times higher than SSD architecture.

For *face detection* the backbone of the Faster R-CNN is an Inception+ResNetV2 network that has the architecture similar to Inception V4. The overall diagram implies a five convolutions block in the first layers, five Inception-ResNetV2-typeA, ten Inception-ResNetV2-typeB and five Inception-ResNetV2-typeC. The type A and type B Inception-RResNet layers were followed by a reduction in feature size, while the type C Inception-ResNet layer was followed by average pooling. The average pooling was performed before fully connected (FC) layers and the dropout was chosen to be 0.8.

#### 2.1.3. Pipeline for Deep Face Detection CNN

Transfer learning and fine-tuning has been used for training SSD and Faster R-CNN models pretrained on COCO and Open Image databases. The input for the fine-tuned training consisted of 2800 face images randomly selected from the Open ImageV4dataset images [50,51]. The dataset was split into three categories: 2000 images for training, 400 for validation and 400 for testing.

The hyperparameters for training each network are described below. The values were obtained under experiments and represent the best configuration in terms of accuracy, generalization and inference speed obtained after training each model. Due to the limitations of the graphical process unit used for training the number of operations was reduced for each training epoch, which implied a low number of images per batch, especially for Faster R-CNN model.

For fine-tuning the SSD architecture, the hyper-parameters were set as follows:Batch size = 16 images;Optimizer: momentum RMS optimizer;Initial learning rate: 6.0 × 10^−5^;Learning rate after 100k steps: 6.0 × 10^−6^;Momentum optimizer: 0.9;Number of epochs: 150k;Dropout: 0.5.

For fine-tuning the Faster R-CNN architecture, the hyperparameters were set as follows:Batch size = 1 images;Optimizer: momentum schedule learning;Initial learning rate: 6.0 × 10^−5^;Momentum optimizer: 0.9;Number of epochs: 150k;Dropout: 0.5.

### 2.2. Facial Emotion Recognition CNN

The database used for training FER CNN is a selection of uncontrolled images from FER2013 and laboratory controlled images from CK+, JaFFE and KDEF [28]. This dataset was divided in 24,336 training, 6957 validation and 3479 test sets. The labeling of the training and testing dataset were previously made by the authors of the database and, in addition, were verified by our team in order to avoid bias. A small amount of the images that did not meet our criteria in terms of class annotations and distinctiveness were dropped or moved to the corresponding class. In addition, data augmentation was used during training in order to increase the generalization of the model. A series of rotations, zooming, width and height shifting, shearing, horizontal flipping and filling were applied to the training dataset. The FER2013 images represents the majority of the dataset, around 90% for each emotion. 

The facial expressions were divided in seven classes: angry, disgust, fear, happy, neutral, sad and surprise. Training classes distribution over the dataset was angry 10%, disgust 2%, fear 3%, happy 26%, neutral 35%, sad 11% and surprise 13%. The validation and test set followed the same distribution. 

The CNN models used for FER were pretrained on the ImageNet database and could recognize objects from 1000 classes. We did not need a SSD or RPN architecture as the face localization was already achieved with face detection CNN. ImageNet did not provide a class related to face or humans but there were some other classes (e.g., t-shirt or bowtie) that helped the network to extract these kinds of features during the prelearning process. This is related to the fact that the network needs these features in order to classify emotions related to classes. Taking into account that only some bottleneck layers will be trained, we would use transfer learning and fine-tuning, as follows: in the first phase the fully connected layers of the pretrained model were replaced with two new randomly initialized FC layers that were able to classify the input images according to our dataset and classes. During this warm-up training all the convolutional layers were frozen allowing the gradient to back-propagate only through the new FC layers. In the second stage the last layers of the convolutional networks were unfrozen, where high-level representations were learned allowing the gradient to back-propagate through these layers but with a very small learning rate in order to allow small changes to the weights. The representation of the three CNN (VGG, ResNet and Inception V3) and the transfer learning approach is shown in Figure 1.

#### 2.2.1. VGG16

In order to avoid over fitting several techniques were taken into account, such as shuffling data during training or adopting dropout. The images were shuffled during training to reduce variance and to make sure that the batches are representative to the overall dataset. On the other hand, using dropout in the fully connected layer reduced the risk of over fitting by improving the generalization. This type of regularization reduced the size of the network by “dropping” an amount of different neurons at each iteration. The CNN was made of five convolutional blocks (16 or 19 layers) each followed by feature map reduction using max-pooling. The bottleneck layer output was passed through an average pooling in order to flatten the feature map to a 1 × 1 × 4096 and the representation was forwarded to the FC network. The FC had one dense layer of 512 neurons with rectified linear unit (ReLu) activation function and dropout of 0.7 and a Softmax classifier.

For fine-tuning the VGG model, the hyperparameter values were obtained under experiments and represent the best configuration in terms of accuracy, generalization and inference speed after training each model and were set as follows:Batch size = 32 images;Optimizer: momentum RMS optimizer;Initial learning rate: 1.0 × 10^−4^;Momentum optimizer: 0.9;Number of epochs: 50, for training the new FC layers;Number of epochs: 50, for training the last convolutional layers;Dropout: 0.5.

#### 2.2.2. ResNet

ResNet is a deep convolutional network that used identity convolutional blocks in order to overcome the problem of vanishing gradients (Figure 2). The gradient may become extremely small as it is back-propagated through a deep network. The identity block use shortcut connections, which are an alternate path for the gradient to flow through, thus solving the problem of vanishing gradient. We would use ResNet with 50 convolutions, which were divided into five stages. Each stage had a convolutional and an identity block, while each block had three convolutions, with 1 × 1, 3 × 3 and 1 × 1 filters, where the 1 × 1 kernel (filter) was responsible for reducing and then increasing (restoring) dimensions. The parameter-free identity shortcuts are particularly important for the bottleneck architectures. If the identity shortcut is replaced with the projection, one can show that the time complexity and model size are doubled, as the shortcut is connected to the two high-dimensional ends. Thereby, identity shortcuts lead to more efficient models for the bottleneck designs [31]. 

Fine-tuning of ResNet architecture, presented at the bottom of Figure 2, implied the following hyperparameter configuration. 

Batch size = 16 images;Optimizer: rectified Adam;Initial learning rate: 1.0 × 10^−4^;Momentum optimizer: 0.9;Number of epochs: 50, for training the new FC layers;Number of epochs: 50, for training the last convolutional layers;Dropout: 0.5.

The values were obtained under experiments and represent the best configuration in terms of accuracy, generalization and inference speed after training each model.

#### 2.2.3. InceptionV3

Inception V3, presented in Figure 3, is a deep neural network with 42 layers, which reduced the representational bottlenecks. It was composed of five stem convolutional layers, three type A Inception blocks, followed by a type A reduction block, four type B Inception blocks and one reduction block, two type C Inception blocks followed by an average pooling layer and the fully connected network. In order to reduce the size of a deep neural network the factorization was taken into account. Different factorization modules were introduced in the convolutional layers to reduce the size of the model in order to avoid over fitting. Neural networks performed better when convolutions did not change the size of the input drastically, reducing the dimensions too much causing loss of information. One factorization implied splitting 5 × 5 convolutions to two 3 × 3 convolution (type A inception block). In addition, factorization of the n × n filter to a combination of 1 × n and n × 1 asymmetric convolutions (type B inception block) was found to dramatically reduce the computation cost. In practice, it was found that employing this factorization does not work well on early layers, but it gives very good results on medium grid-sizes [52]. The last factorization taken into account was the high dimensional representations by replacing two of the 3 × 3 convolution with asymmetric convolutions of 1 × 3 and 3 × 1. 

For fine-tuning the Inception V3 model, the hyperparameters configuration is as follows:Batch size = 32 images;Optimizer: Adam;Initial learning rate: 1.0 × 10^−4^;Momentum optimizer: 0.9;Number of epochs: 50, for training the new FC layers;Number of epochs: 50, for training the last convolutional layers;Dropout: 0.5.

The values were obtained under experiments and represent the best configuration in terms of accuracy, generalization and inference speed after training each model.

### 2.3. Optimization Using Rectified Adam and Batch Normalization

There are several methods that accelerate deep learning model optimization by applying adaptive learning rate, such as the adaptive gradient algorithm (Adagrad), Adadelta, Adamax, root mean square propagation (RMSprop), adaptive moment optimization (Adam) or Nesterov adaptive moment optimization (Nadam). Rectified Adam is a state-of-the-art version of the Adam optimizer, developed by [53], which improves generalization and introduces a term to rectify the variance of the adaptive learning rate, by applying warm up with a low initial learning rate.

Computing the weights according to the Adam optimizer:(1)$$w_{t}=w_{t-1}-\eta \frac{{\hat{m}}_{t}}{\sqrt{{\hat{v}}_{t}}+\varepsilon }$$

The first moving momentum:(2)$$m_{t}=(1-\beta_{1})\sum_{i=0}^{t}\beta_{1}^{t-i}g_{i}$$

The second moving momentum:(3)$$v_{t}=(1-\beta_{2})\sum_{i=0}^{t}\beta_{2}^{t-i}g_{i}{}^{2}$$

The bias correction of the momentums:(4)$${\hat{m}}_{t}=\frac{m_{t}}{1-\beta_{1}^{t}}$$
(5)$${\hat{v}}_{t}=\frac{v_{t}}{1-\beta_{2}^{t}}$$

Adding the rectification term in Equation (1), the recent variant of Adam optimization, named rectified Adam (RAdam), has the form:(6)$$w_{t}=w_{t-1}-\eta r_{t}\frac{{\hat{m}}_{t}}{\sqrt{{\hat{v}}_{t}}}$$
where the step size, η, is an adjustable hyperparameter and rectification rate is:(7)$$r_{t}=\sqrt{\frac{(p_{t}-4)(p_{t}-2)p_{\infty }}{(p_{\infty }-4)(p_{\infty }-4)p_{t}}}$$
while $p_{t}=p_{\infty }-\frac{2t\beta_{2}^{t}}{1-\beta_{2}^{t}}$ and $p_{\infty }=\frac{2}{\left(1-\beta_{2}^{t}\right)}-1$.

When the length of the approximated simple moving average is less or equal than 4, the variance of the adaptive learning rate is deactivated. Otherwise, the variance rectification term is calculated and parameters are updated with the adaptive learning rate.

After applying batch normalization to the activation function output of the convolutional layers, the normalized output will be:(8)$$x_{i}=\gamma {\hat{w}}_{i}+\beta$$
where γ and β are parameters used for scale and shift that are learned during training. Moreover, the weights normalization over a mini-batch is:(9)$${\hat{w}}_{i}=\frac{w_{i}-\mu_{B}}{\sqrt{\sigma_{B}^{2}+\varepsilon }}$$
where the mini- batch average:(10)$$\mu_{B}=\frac{1}{m}\sum_{i=1}^{m}w_{i}$$
and the mini-batch variance:(11)$$\sigma_{B}^{2}=\frac{1}{m}\sum_{i=1}^{m}{\left(w_{i}-\mu_{B}\right)}^{2}$$

For FER our end-to-end human–robot interaction pipeline used convolutional neural network models that were trained using batch normalization after ReLu activation and RAdam optimizer. As we will later see in the paper the best results using RAdam have been obtained for the ResNet model.

### 2.4. Experiment Setup

In order to develop an end-to-end pipeline for the interaction between human and NAO robot using computer vision based on deep convolutional neural networks a preprocessing CNN for facial detection was added before the FER CNN.

The entire facial expression pipeline was implemented on the NAO robot and is presented in Figure 4. The system was divided in four parts: the NAO robot image caption (NAO camera), the face recognition model (FR CNN) and the facial emotion recognition model (FER CNN) and robot facial expression (output to the user/human). For image caption and output to the user we used the Naoqi library functions running on the robot, while the FR and FER models were uploaded to the robot and enabled when the emotion recognition was activated. 

NAO is a humanoid robot developed by Aldebaran (Softbank Robotics), it has a height of 57 cm, weighs 4.3 kg and is widely used in the research and education due to its good performances, small size, affordable price and the wide range of sensors it is equipped with. Since the scripts can be developed in several programming languages and can be compiled both locally (on the robot) and remotely, NAO can be used in various applications. In order to achieve a human–robot interaction NAO’s top front camera was used for taking pictures or videos of the person in front and NAO’s voice module together with eye and ear LEDs for outputting robot emotion. NAO has two identical RGB video cameras located in the forehead, which provide a 640 × 480 resolution at 30 frames per second. The field of view is 72.6°DFOV with a focus range between 30 cm and infinity [54]. Eye and foot LEDs are RGB full color, so it is possible to combine the three primary colors (red, green and blue) in order to obtain different colors. This feature will be used to associate an emotion to one color: happy is green, red is angry, blue is sad, disgust is yellow, neutral is black, surprise in white and fear is orange (Figure 5). The intensity of the color will be adjusted depending on the probability of emotion detection.

In terms of computer vision, NAO’s embedded module is able to detect faces with a miss rate of 19.16% for frontal head positions, based on a face detection solution developed by OMRON. The NAO robot software (Choregraph) and libraries share a computer vision module for facial recognition named ALFaceDetection, which is able to recognize one particular face previously learned and stored in its limited flash memory. The learning and storing process of one face is tedious and involves several steps, in which, for example, NAO checks that the face is correctly exposed (no backlighting, no partial shadows) in three consecutive images. Compared to the NAO face detection module, the CNN based facial recognition, which represents the preprocessing module of the pipeline presented in the paper, was straightforward and could recognize one random face with an accuracy of 97.8% as it will be presented in the Section 3.1.1.

The training of the FR and FER models were performed on a GeForce GTX 1080 graphics processing unit (GPU) running on compute unified device architecture (CUDA) and CUDA deep neural network library (CUDNN) with the following specifications: memory 8 GB GDDR5X, processor 1.8 GHz and 2560 CUDA cores. The inference (detection) was running on the NAO robot, which was equipped with an Intel Atom Z530 1.6 GHz CPU and 1 GB RAM memory.

In order to increase the inference time we used a neural network preprocessor developed by Intel, called the Neural Compute Stick 2 (NCS 2), together with asynchronous threading, which boosts the NAO performances from 0.25 frames per second (FPS) to 4-6 FPS on our FR+FER pipeline. The inference speed when using only one CNN, reached 8-9 FPS with NCS 2.

## 3. Results

### 3.1. Facial Recognition 

#### 3.1.1. ResNet-Inception Based Faster R-CNN Model 

The loss variation of Faster R-CNN is presented in Figure 6. It can be seen that the loss dropped very quickly. This happened because the complex architecture of R-CNN, which is a combination of ResNet and Inception models, increased the training accuracy. Another parameter that artificially boosted the accuracy of the network was the number of images in the batch. In our case there was only one image per batch due to the limitations of the GPU. Since the gradient descent was applied to every batch, not to the entire dataset, it was normal to have a rapid drop and small loss overall. After the rapid drop to a value of 0.4 in the first 5k steps, the loss decreased slowly until it started to converge at around 60k steps with the value of 0.25. The accuracy of Faster R-CNN model was 97.8% using the Pascal visual object class (VOC) evaluation metrics.

#### 3.1.2. Inception Based SSD Model 

In contrast with Faster R-CNN the smoothened loss for SSD Inception has an exponential dropping and converged at around 120k steps. The loss variation can be observed in Figure 7, with the minimum value around 1.8. The precision of SSD Inception model was 97.42% using the Pascal VOC evaluation metrics.

### 3.2. Facial Emotion Recognition

#### 3.2.1. VGG16

The fine-tuning of the FC layers and training of the last convolutional layers of the VGG model for FER are presented in Figure 8. The training (red line) and validation loss (blue line) during the “warm-up” drop after several epochs and, after completing the FC layers training, the train loss was 1.1 and validation loss was 0.8 (Figure 8a). The accuracy was 0.6 (60%) for training (purple line) and 0.7 (70%) for validation (gray line). Once the classification layers weight was initialized according to new output classes and learns some patterns from the new dataset, it was possible to skip to the next step. The last two convolutional layers and the max pooling of the VGG model were unfrozen in order to get feature maps that relate to emotion recognition. Initially, the training was set for 100 epochs but because the model started to over fit, the learning process was stopped after 50 epochs. When the training completed, the train loss (red line) dropped to 0.36 and the validation loss (blue line) started to become instable and converge around 0.5. The accuracy of training (purple line) increased to 0.87 (87%) while the loss (gray line) reached 0.84 (84%; Figure 8b).

#### 3.2.2. ResNet

In Figure 9 the fine-tuning of the fully connected layers and last unfrozen layers of ResNet model is presented. The fully connected network was made of one dense layer of 1024 neurons activated by a ReLu function and one dense layer for classification activated by a Softmax function. The model that achieved the best results was compiled using a rectified Adam optimizer with a learning rate value set at 0.0001 

The training of these two layers was made while keeping all the model layers frozen. The losses of the training and validation set converged around the value of 1.1 and the accuracy of both sets around 0.6 (Figure 9a). For the actual training of the entire model we tried different fine-tunings of the hyperparameters while unfreezing different top layers of the ResNet architecture. The setting that achieved the best results was: batch size: 16, optimizer rectified Adam, learning rate—0.0001 and unfreezing the layers from the 31st convolution (the three ID blocks before the last convolutional block). This model reached a training accuracy of 0.9014 (90.14%) when the learning process was stopped, after 50 epoch, because the validation loss started to increase and model was over fitting (Figure 9b). In other tests the training accuracy increased to 0.96 while over fitting the model but for our FER inference system we want the network to have a good generalization on different datasets. The ResNet gave the best results of all tested FER architectures, in terms of generalization and was used for the NAO-SCNN inference system pipeline.

#### 3.2.3. InceptionV3

The results of fine-tuning the InceptionV3 network are presented in Figure 10. As with the VGG and ResNet models the training was divided in two steps: first the weights of the fully connected layers were updated while freezing the rest of the network and then some of the layers of the Inception B block together with the Reduction B and Inception C block were unfrozen (Figure 10a). The fully connected network was made of one dense layer with 512 neurons activated by a ReLu function and one dense layer for classification activated by a Softmax function. The model was compiled using rectified Adam, Adam, RMSProp and stochastic gradient descent (SGD) optimizers with a learning rate value set at 0.0001. The training of the entire network was made by combining different hyperparameters values while unfreezing different top layers of the InceptionV3 architecture. 

The best results were achieved for batch size: 32, Adam optimizer, learning rate—0.0001 and unfreezing the layers starting from the fifth convolution of the first Inception Block-type B. This model reached a training accuracy 0.81 (81%) and 0.78 (78%) when the training was stopped, after 50 epoch (Figure 10b), due to over fitting.

The confusion matrix for FER predictions of the three architectures is presented in Figure 11. The network with the best overall results, from the confusion matrix perspective, was also ResNet (Figure 11b), followed by the VGG (Figure 11a) and InceptionV3 (Figure 11c). The first thing to notice was that the class with the highest score was the “happy” class regardless of the network model. There were two crucial factors that enabled such a good prediction: high variance and large dataset, with the first being the most important. Training samples play an important role for learning a complex future but the variance was determinant when it came to highlighting crucial features. This can be seen from the fact that the happy class is not the biggest in the training samples, being overtaken by the neutral class. Although it had the largest share of the dataset, the “neutral” class failed to generalize as well as others. The same thing when comparing angry, sad and surprise classes. The classes shared the same proportion from the dataset but the accuracy was different.

All of these observations mentioned above were influenced by emotion variance. The similarity of the emotion variation could also be observed in the confusion matrix. For example “fear” can be easily mistaken with “surprise” and “sad” with “neutral. An important number of images from “angry” and “sad” classes were classified as “neutral” images. This happens because these classes had low variance, in terms of mouth and eyebrows shape. The shape of mouth did not change significantly, while the displacement of the eyebrows was hard to be distinguished by the CNN model. It is interesting that the vice versa was not happening, probably because of the high share of training samples. Another important misclassification was the fear emotion classified as surprise. Due to the similarity of these emotions in terms of mouth shape and low changes in the eyebrows shape, fear was overly misclassified as surprise (19%). The misclassification of the classes could be improved by increasing the number of images and by removing the neutral class in order to force the model to learn other distinct features. Bias played an important role, mainly when the expressions were difficult to be labeled, thereby affecting the models that were trained on them. 

In Table 1, the performances of state-of-the-art CNN models trained on FER2013 database are presented with respect to the CNN type, preprocessing method and optimizer. The NAO-SCNN models developed in this research achieved the highest accuracy, with the ResNet based architecture obtaining the best performances.

In Table 2, the performances of state-of-the-art CNN models used for emotion recognition on NAO robot are presented. The NAO-SCNN models achieved the highest accuracy, with the ResNet based architecture obtaining the best performances.

In Figure 12 the manner the VGG model learns patterns for a given input image is presented. We followed the feature maps transformation of two outputs from convolutional blocks four and nine. After the convolution four our fine-tuned model focused on learning different edges like face, mouth, nose, eyes or teeth outline. This was the result of the convolutional filters learning how to identify lines, colors, circles or other basic information. 

The feature maps after the second max pooling were concentrated to learn more complex representations from the picture. At this stage the convolutional filters were able to detect basic shapes. The filters from the deeper layers were able to distinguish between difficult shapes at the end of training and the representation of the feature maps became more abstract.

The feature maps for two layers of ResNet model are presented in Figure 13. The first set of images were captured after the second convolutional block. At this point, the feature maps had high resolution and focused on sparse areas of the face. The second set of images was taken from the second convolution of third ID block. It could be observed that at this point the filters were concentrated on particular areas, such as eyes, mouth, teeth, eyebrows and the representation of the feature maps was more abstract.

The feature maps for two layers of Inception V3 model are presented in Figure 14. The first set of images was captured after the first Inception type-A block. As VGG and ResNet, the convolutional filters were able to detect the shape of areas responsible for emotion detection such as eyes, mouth, eyebrows and also neighbor regions with relevant variance.

The second set of images was taken after the first Inception block type B. At this point, the resolution was low as the filters concentrated on small regions of the input image and the feature maps had a more abstract representation.

## 4. Conclusions

The SSD and the Faster R-CNN models for face detection shared almost the same total loss and accuracy. The accuracy of the Faster R-CNN model was 97.8% based on the VOC metrics, while for the SSD Inception model it was 97.42%. Thus, taking into account that Faster R-CNN is a large network, in terms of total parameters, the SSD architecture was chosen for FR detection in order to keep a low inference speed.

The network with the best overall performances, in terms of accuracy and loss for FER detection was ResNet and the results were confirmed by the confusion matrix. This model obtained a training accuracy of 90.14%, while VGG had 87% accuracy and Inception V3 reached 81%. By adding 5–6% of laboratory controlled images from other databases the accuracy of FER2013 increased with more than 10%, of the highest score so far, using the ResNet model together with other optimizations. In order to develop a model that better generalizes FER it is important to mix controlled and uncontrolled images.

We also noticed that using the preprocessing module for FR before running the inference for emotion recognition enhances the FER confidence score for the same images. The results show improvements well over 10% when using two serialized CNN, instead of just using the FER CNN. 

A recent optimization model, called rectified Adam, was applied for training the three FER CNN models, which led to a better generalization. RAdam introduces a term to rectify the variance of the adaptive learning rate, by applying warm up with a low initial learning rate. The tests showed a more robust loss during training and accuracy improvement of 3%-4% on each FER CNN compared to other optimizers.

The FR and FER architectures were implemented on the NAO humanoid robot in order to switch from the current computer vision library running on the robot to a newer and more accurate serialized model based on deep convolutional neural networks. 

In order to increase the inference time we used a neural network preprocessor developed by Intel, called Neural Compute Stick 2(NCS 2), together with asynchronous threading, which boosts the NAO performances from 0.25 to 4-6 FPS on our FR+FER pipeline. The inference speed when using only one CNN, reached 8-9 FPS with NCS2.

Future developments will involve the fusion of other inputs, such as audio models, infrared images, depth information from 3D face models, physiological data, etc. will provide additional information and further enhance the accuracy of the process.

The human robot interaction model obtained during this research will be further developed in order to be applied for medical purposes, mainly for improving communication and behavior of children with autism spectrum disorder. The research will be carried out using medical feedback from psychologists, physicians, educators and therapists to reduce frustration and anxiety through communication with a humanoid robot.

## Figures and Tables

**Figure 1 sensors-20-02393-f001:**
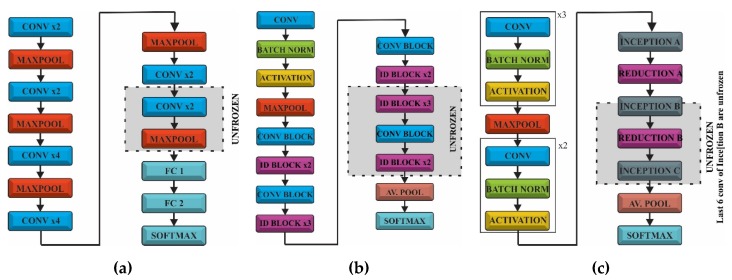
Architectures of the models used for facial emotion recognition (FER): (**a**) VGG; (**b**) ResNet50 and (**c**) Inception V3.

**Figure 2 sensors-20-02393-f002:**
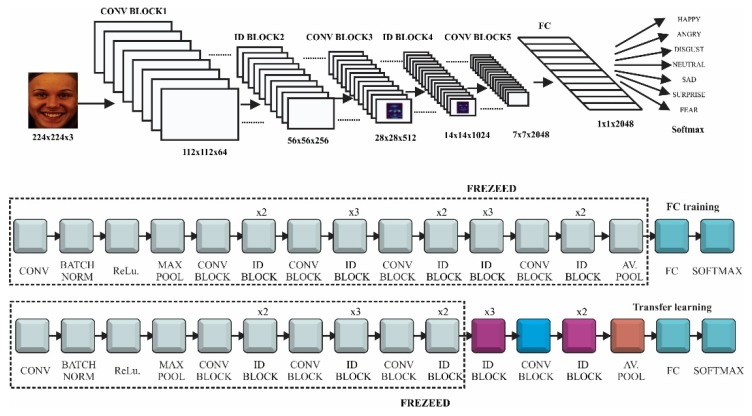
ResNet model using transfer learning and fine-tuning.

**Figure 3 sensors-20-02393-f003:**
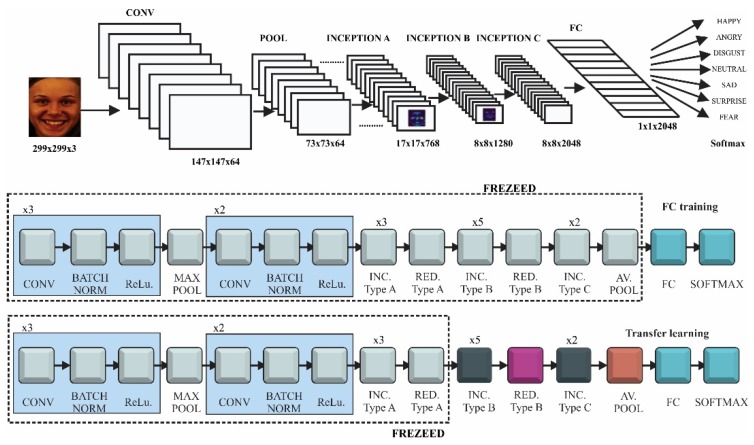
Inception model using transfer learning and fine-tuning.

**Figure 4 sensors-20-02393-f004:**
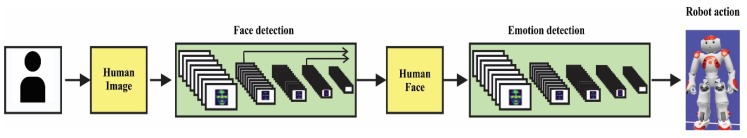
End-to-end emotion detection pipeline.

**Figure 5 sensors-20-02393-f005:**
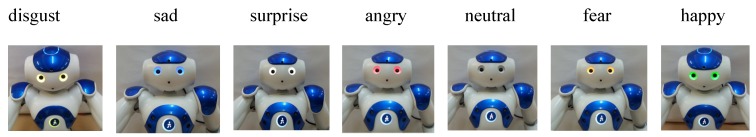
NAO color expression.

**Figure 6 sensors-20-02393-f006:**
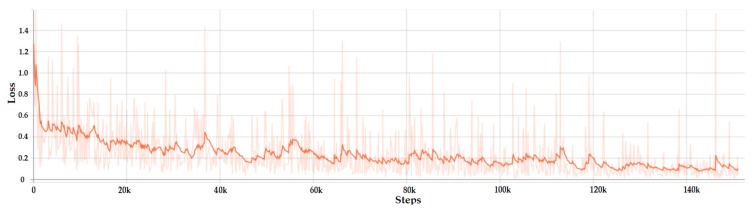
Total loss of the faster region-based convolutional neural network (Faster R-CNN) model.

**Figure 7 sensors-20-02393-f007:**
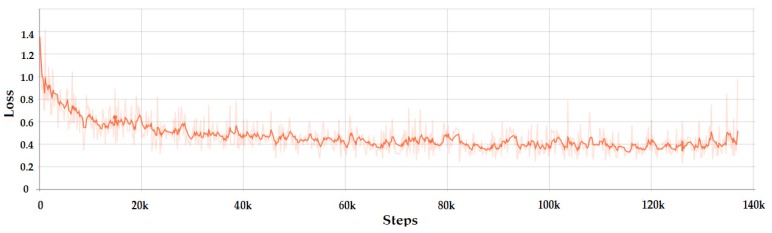
Total loss of the Inception based single shot detector (SSD) model.

**Figure 8 sensors-20-02393-f008:**
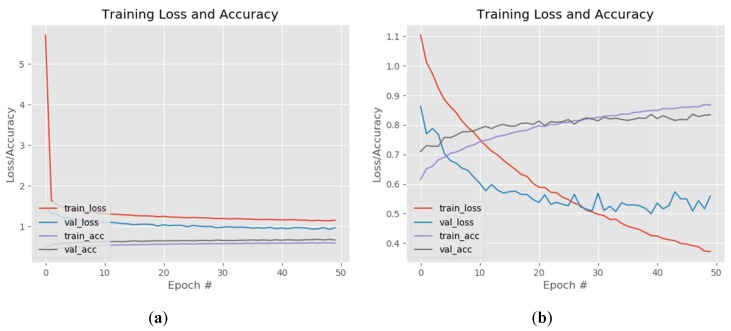
VGG training loss and accuracy: (**a**) fully connected (FC) layers warm-up and (**b**) training of last two convolutional layers.

**Figure 9 sensors-20-02393-f009:**
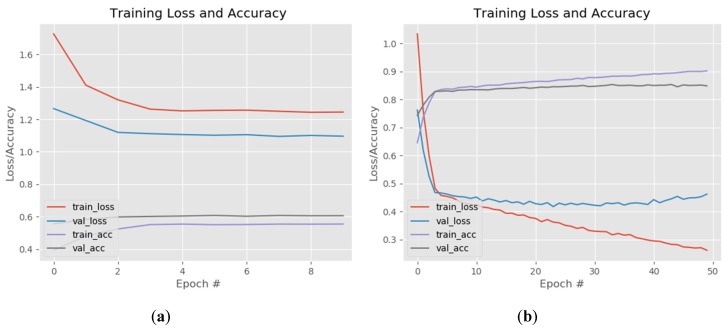
ResNet training loss and accuracy of the FC layers warm-up (**a**) and training of the last five convolutional layers (**b**).

**Figure 10 sensors-20-02393-f010:**
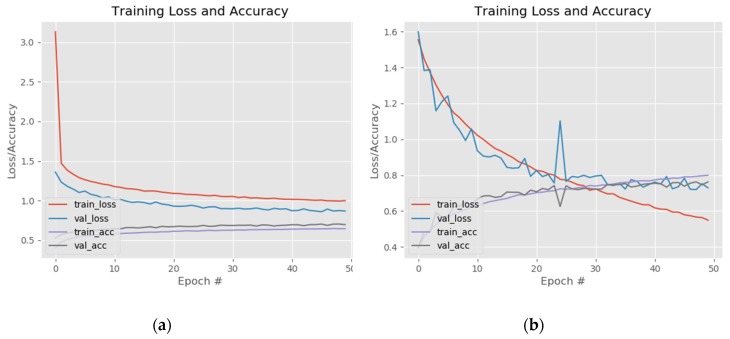
InceptionV3 training loss and accuracy for: (**a**) FC layers warm-up and (**b**) training of the last five convolutional layers.

**Figure 11 sensors-20-02393-f011:**
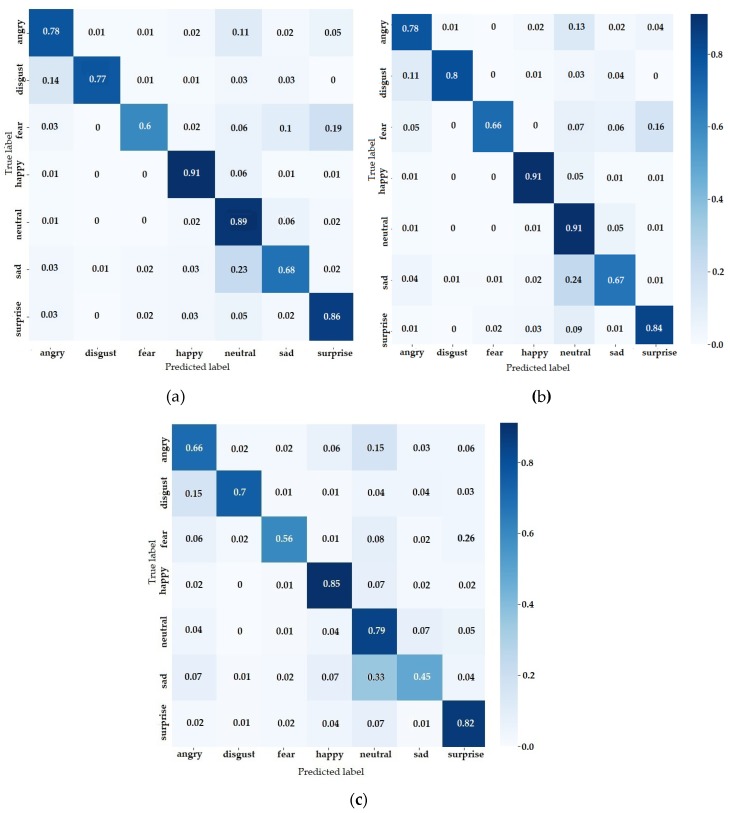
Confusion matrix for: (**a**) VGG; (**b**) ResNet and (**c**) InceptionV3.

**Figure 12 sensors-20-02393-f012:**
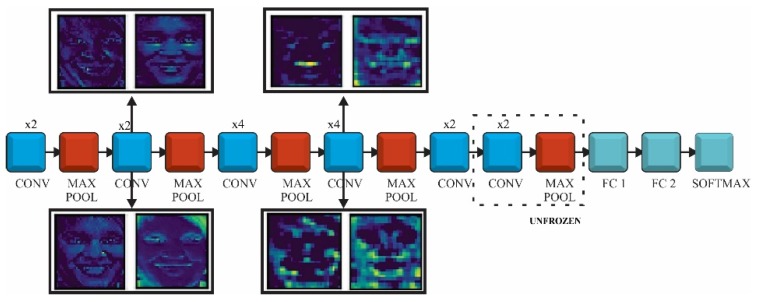
VGG feature maps at convolution 4 and convolution 9.

**Figure 13 sensors-20-02393-f013:**
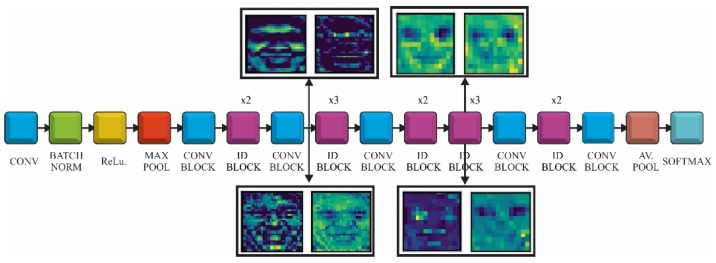
ResNet feature maps after the second convolution block and 8th ID block.

**Figure 14 sensors-20-02393-f014:**
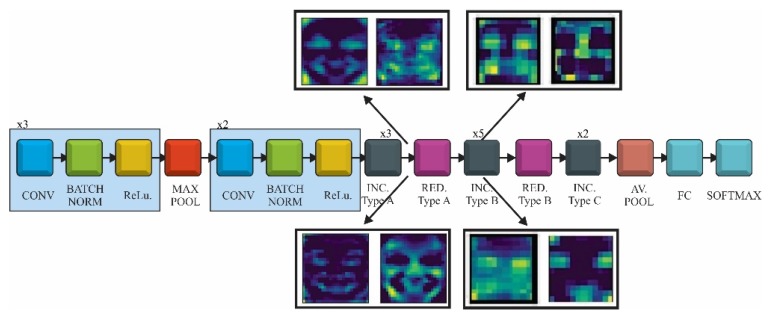
Inception V3 feature maps after the Inception block type A and Inception block type B.

**Table 1 sensors-20-02393-t001:** Results for models trained on FER2013 database.

CNN Model	Accuracy [%]	CNN Type	Preprocessing	Optimizer
Pramerdorfer et al. [22]	75.2	Network Ensemble	-	SGD
Tang [23]	71.2	Loss layer	-	SGD
Kim et al. [24]	73.73	Network Ensemble	Intraface	SGD
Minae et al. [25]	70.02	Loss layer		Adam
Hua et al. [26]	71.91	Ensemble Network	Straightforward	Adam
Connie [27]	73.4	Multitask Network	-	Adam
**NAO-SCNN (ResNet)**	**90.14**	**Loss layer**	**CNN**	**RAdam**
NAO-SCNN (Inception)	81	Loss layer	CNN	Adam
NAO-SCNN (VGG)	87	Loss layer	CNN	Adam

**Table 2 sensors-20-02393-t002:** Results for models used on NAO robots.

CNN Model	Accuracy [%]
CNN Viola-Jones [1]	44.88
DBMM Classifier [2]	85
NAO-SCNN (ResNet)	90.14
NAO-SCNN (Inception)	81
NAO-SCNN (VGG)	87

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
