# Peer review of "Facial Expressions Recognition for Human–Robot Interaction Using Deep Convolutional Neural Networks with Rectified Adam Optimizer"

_sensors, 2020, doi:10.3390/s20082393_

Round 1
Reviewer 1 Report
This paper tries to use different CNN models to interpret the facial expression of human to achieve the interaction with a NAO robot. However, there are some serious problems need to be resolved before consider for publication.
- the authors mainly talked about using CNN models to recognize face and facial expression based on human pictures, but barely talked about the “Interaction” between human and NAO robot. After recognizing the expression of human, what is the response of robot? Just you happy, I happy (Green), you sad I sad (blue)?
- The results in this paper only showed the evaluation results of CNN models, no much description about the results of face and facial expression recognition. Which platform all the calculations in this paper performed on?
- Nao robot is a tremendous programming tool and it has especially become a good platform for education and research. It is believed that the robot’s recognition system for human face and facial expression have already been good enough for responding human reaction. Then what is the point of the research in this paper? What is the advantage of this research compared with the system NAO robot already have? The authors need to point them out clearly.
Some minor mistakes
- there are so many abbreviations used in this paper, some of them have no full name listed when they first appeared, such as line 19, VOC? SSD? The authors need to check them all over again carefully.
- Some sentences are not properly written. Such as line 22, “….., instead of just using the FER CNN”, the author must mean “instead of just using one FER CNN model”.
- Some figures in this paper is in the trouble of low resolution, such as figure 6-11, the title size of axis need to be larger.
- line 413, 3.1.3. InceptionV3? 3.2.3. Inception V3
Author Response
Dear Editor and Reviewer,
Thank you for your letter and for the reviewers’ comments concerning our manuscript. The authors express their gratitude to all reviewers for their insightful comments.
Those comments are all valuable and very helpful for revising and improving our paper, as well as important in guiding the significance of our research.
The authors' answer is attached in the Word file.

Reviewer 2 Report
In the lines 77 through 79, Clarify, they are databases or methods?
I suggest to orginize in a Table all the algorithms, databases, methods, detectors with advantages and disadvantages and all the other characteristics around them to facilitate a better comprehension of all methods, algorithms and databases used joint to the acronyms used to identify different situations of the phenomenons described
Care all the acronyms used to describe different treatments, for example, SSD-Single shot detector is the same case used to describe the FR in the model SSD-CNN?. I recommend the authors standardize all the definitions.
In the section, the authors don´t talk about the face recognition importance, it is important to include some paragraphs explaining the reasoning behind this topic.
Care about the acronyms used, standarize the use of them, i. e. Fast R-CNN or Fast RCNN, because it tends to confuse the reader (line 156 and 161)
In the phrase "The overall diagram implies a five convolutions block in the first layers, five Inception-ResnetV2-typeA, ten Inception-ResnetV2-typeB and five Inception-ResnetV2-typeC." --- Procedure given in fact or proposal of the authors?
In the lines 185, 194, 241, 263, 290, Justify the values of the parameters used. These values were obtained under experiments?
In the lines 209 and 210, the training clases distributions were previously labeled?, how or under which criteria they were labeled?
in the line 319, Initialization values must be given and don't appear in the paper, please justify.
In general, must take care about grammatics in the whole paper, there exist a lot of mistakes, i. e., in the lines 323 "neural etwork", 324"normalizarion afer", 330 "is introduces", etc., please check the document.
in line 391, please clarify, I thought this number is wrong agreed to the Fig 8b), I think is "accuracy validation" instead of "loss reached"
Author Response

(The authors gave the same response as above.)

Reviewer 3 Report
This paper proposes a face expression recognition system base on CNNs. The method used existing deep neural network architectures and traditional rectified Adam optimizers.
The main drawback of this paper is lack of novelty. The method employed existing techniques for face expression recognition. The proposed system is also straightforward to implement.
The experiments are insufficient to demonstrate the effectiveness of the method. The proposed method has to be compared with recent state-of-the-art algorithms.
Author Response

(The authors gave the same response as above.)

Reviewer 4 Report
This manuscript varified the effectiveness of several typical deep learning models for facial expression recognition for human-robot interaction. It is very interesting. The experiments are well designed and the results are well presented. There are a few points to be improved.
- The title is not appropriate and should be improved. Artificial Intelligence is a broad concept, and in your work only some deep learning models are used.
- Several figures are not clear. Please redraw.
Author Response

(The authors gave the same response as above.)

Round 2
Reviewer 3 Report
All issues raised by reviewers were carefully addressed in the revised manuscript. Now a current manuscript is ready for the publication.